# Study on Urban Land Ecological Security Pattern and Obstacle Factors in the Beijing–Tianjin–Hebei Region

**Wenying Peng \*, Yue Sun, Can Liu and Dandan Liu**

School of Urban Economics and Public Administration, Capital University of Economics and Business, Beijing 100070, China
* Correspondence: cjpengwy@cueb.edu.cn

**Abstract:** Land ecological security is the material basis of the sustainable development of human society. The coordinated development of the Beijing–Tianjin–Hebei region is a major national strategy of China. Land ecological security is of great significance to the coordinated development of the Beijing–Tianjin–Hebei region and the maintenance of China's ecological security. In this paper, the pressure–state–response (PSR) model is used to construct an evaluation index system of land ecological security, an entropy-weight technique for order preference by similarity to ideal solution (TOPSIS) method is used to calculate the land ecological security index in the Beijing–Tianjin–Hebei region, and an obstacle degree model is used to reveal the obstacle factors. The results show that the overall level of land ecological security in the Beijing–Tianjin–Hebei region was low, and that the security level presented a pattern of "high in the north and low in the south". The land ecological security level was mainly affected by the state subsystem and response subsystem, and the average index of the pressure subsystem was 0.543, which reached the safe state. The main obstacle factors are per capita grassland area, per capita forest area, green land rate of built-up area, urbanization rate, per capita cultivated land, etc. This study provides a theoretical basis for the construction of the land ecological security system, sustainable utilization of land resources and regional sustainable development in the Beijing–Tianjin–Hebei region and promotes the formation of a benign circulation pattern of land ecosystem and effective prevention and control of land ecological and environmental risks in the region.

**Keywords:** land use; ecosystem; ecological security; Beijing–Tianjin–Hebei region

## 1. Introduction

Human activities have been increasing their impact on land cover, disturbing or even damaging the ecosystem, and seriously threatening ecological security [1]. With the development of global industrialization and urbanization, ecological security issues such as environmental pollution, resource shortage, biodiversity decline, soil erosion and desertification have become increasingly prominent. The international community is paying more and more attention to sustainable development and ecological security issues. The Sustainable Development Goals of the United Nations clearly point out that quantitative ecological security is of great significance for the realization of regional sustainable development. The concept of ecological security–including natural security, economic security, social ecological security and human welfare–emphasizes the state of ecological environment necessary for the safe development of a society or country [2]. Existing studies have found that the sensitivity of the eco-environmental system is affected by the degree of disturbance and damage of human production activities and natural disasters on land resources and the environment, and the degree of disturbance and damage and the bearing limit of the environment are difficult to accurately measure [3]. Land is a complex of various natural elements including geology, geomorphology, climate, hydrology, soil, biology and other factors as well as the effects of human social production and life. Furthermore, land is

the carrier of terrestrial ecology and the natural basis for human survival and development [4]. Land ecological security is the basic guarantee of ecological security and an important part of natural ecological security [5]. Land ecological security refers to the dynamic balance between natural elements and human activities on the land surface [6], and the balance between internal and external land systems [7]. Land ecological security is closely related to human survival and development and is the core and foundation of sustainable utilization of land resources. In September 2015, the United Nations adopted the 2030 Agenda for Sustainable Development, which includes 17 sustainable development goals and 169 targets. The land use goals are basically consistent with the sustainable development goals. The goal of land use is to utilize land reasonably and efficiently, ensure land quality and provide services, and maximize the benefits of products. Additionally, it seeks to improve the quality of human life as far as possible while not exceeding the natural carrying capacity of the land. Maintaining the stability of the structure and function of the land ecosystem is of great significance for the realization of the sustainable development of society and the economy, ensuring the long-term stability and coordinated development of the land ecosystem [8]. However, with the enhancement of human development and construction activities, irrational land use has seriously threatened the health and safety of the ecosystem, resulting in a series of environmental problems such as biodiversity reduction, soil erosion and land pollution [9]. Land ecological security has gradually become a research hotspot. In the important period, where we seek to promote the 2030 Agenda for Sustainable Development, it is of great importance that we adhere to the harmonious coexistence between man and nature and carry out in-depth research on the pattern of land ecological security and its obstacle factors.

Early studies on land ecological security has focused on land ecosystem health theory [10], land ecosystem health evaluation [11], and land ecological security early warning. In the world, ecological security early warning research mainly focuses on the aspects of land quality, land fertility and soil degradation. With the rise of quantitative research on ecosystem sensitivity, environmental capacity and carrying capacity, quantitative evaluation of land ecological security has become one of the research hotspots. The pressure–state–response (PSR) [12] model, developed by the Organization for Economic Cooperation and Development (OECD), can directly express the causes, results and countermeasures of the degradation of ecological environment quality, and accurately reflect the internal relationship of land ecosystem. The PSR evaluation model is one of the most widely used index systems at present and is regarded as the most effective framework for the organization of environmental indicators and the reporting of environmental status. In recent years, the PSR model has been widely used in land quality index system research, agricultural sustainable development system research, environmental protection investment analysis and other fields. With the deepening of ecological security research, the PSR model has been widely used in ecological security assessment in various fields and regions. Xu [13] used the PSR model to select ten evaluation indexes from three aspects, namely pressure, state and response, and built the Linghekou wetland ecosystem health evaluation index system. Du [14] constructed an ecosystem health evaluation system to evaluate the ecosystem health degree of an oasis–desert ecotone in Minqin County; Ji [15] used the PSR model and ecosystem health theory to evaluate the ecological status of coral reefs, and established a coral reef ecosystem evaluation method suitable for China. In recent years, Chinese scholars have paid much attention to the evaluation of land ecological security. Guo [16] established the evaluation index system of the Fen River Basin's ecological security based on the PSR model, calculated the correlation degree of each index combined with the matter–element model, and evaluated the land ecological security quality of the Fen River Basin from a more objective perspective. Chen [17] constructed an evaluation index system of the entropy-weight matter-element model, and conducted research and evaluation on land ecological security in the Jiangjin district of Chongqing from three perspectives: natural conditions, land use status, and socio-economic development level. Zhang [18] applied the ecological footprint method to evaluate the ecological carrying capacity of

12 provinces in western China. Many scholars have applied the "pressure–state–response" (PSR) model [19–21] using an entropy-weight technique for order preference by similarity to ideal solution (TOPSIS) method to calculate the safety index, and divided the ecological security levels according to the equal spacing or non-equal spacing method, so as to explore the ecological security pattern. Shi [22] evaluated the suitability of the ecological security pattern in Huaibei city and divided the ecological security pattern into a low ecological security pattern, high ecological security pattern and medium ecological security pattern according to the comprehensive suitability of land. Su [23] studied land ecological security in the Fen River Basin and found that spatial ecological security was characterized by "large aggregation and small dispersion". Zhang [24] analyzed the land ecological security pattern of major agricultural producing areas and divided it into four parts: land ecological source, ecological security zone, ecological corridor and ecological node. Yue [25] formulated and implemented technical regulations for optimizing a land use structure based on remote sensing image interpretation and GIS technology to evaluate the ecological security of sandy land. The PSR model can better reflect the overall level of the pressure, state and response of the ecosystem brought by human activities in the level of index system construction and evaluation objectives, but it lacks accurate judgment of the impact factors. Therefore, some scholars have applied the obstacle degree model to judge and identify the obstacle factors affecting land ecological security. Fan [26] used the obstacle degree model to determine that economic development and agricultural production activities were the main obstacle factors affecting ecological security; Li [27] analyzed the obstacles to sustainable utilization in the Yellow River Delta and found that gross agricultural output value per capita, grain yield per unit area, reed land occupancy rate and agricultural land occupancy rate were the main influencing factors for sustainable land use in the Yellow River Delta. The obstacle degree model was of guiding significance for the analysis of land ecological security and the optimization of regional ecological security pattern. This study combines two methods, entropy weight and TOPSIS. Since TOPSIS has no special requirements on sample data and can clearly reflect the gap between schemes, it can be used for multi-scheme and multi-objective decision-making. Before using TOPSIS, it is necessary to determine the weight coefficient of each evaluation index. As an objective assignment method, entropy-weight method can more accurately reflect the dispersion degree of each index. Therefore, this paper adopts a combination of the entropy method and the TOPSIS method to avoid the non-scientific and non-objective problems caused by subjective weighting and calculate the weight of each index more accurately.

The coordinated development of the Beijing–Tianjin–Hebei region is a major national strategy of China. It will serve as a model for the integrated and coordinated economic development of global urban agglomerations and the joint development, sharing and governance of ecological environment. In recent years, many studies have focused on land ecosystem assessment in the Beijing–Tianjin–Hebei region. Among these studies, some scholars have revealed the interference process incurred by the process of urbanization in the Beijing–Tianjin–Hebei region on vegetation cover and ecological quality. Han et al. evaluated the urban ecological security in the Beijing–Tianjin–Hebei region from 2003 to 2012 based on the PSR model. Some scholars have also analyzed the evolution law of ecological carrying capacity, ecological deficit and ecological security in the Beijing–Tianjin–Hebei region. Chen [28] classified land ecological security problems into two categories: one is the decline of land productivity, such as water erosion, wind erosion, soil salinization, soil structure damage, soil acidification and fertility decline, and the other is the decline of ecological landscape stability, such as tree wilt, biodiversity reduction and frequent natural disasters. Leopoid also listed several phenomena that targeted the health of the land ecosystem, such as soil erosion, invasive species, loss of indigenous species, and loss of wildlife. Existing studies have used DPSIR–DEA model to construct an assessment model of ecological pressure in the Beijing–Tianjin–Hebei region, and discussed the collaborative regulation of land ecological pressure from the perspective of system coordination [29]. Aiming at the problems of low environmental carrying capacity, heavy non-point source pollution load

and declining ecological environment quality in the Luan River and Chao River basins, the PSR model was used to evaluate the impact of human activities on ecological security in the Beijing–Tianjin–Hebei region from the watershed scale, in which the pressure layer represents the pressure caused by human production and life on the environment. The state layer represents the state of ecological environment, and the response layer represents the means and measures taken by human beings in the face of pressure, including economic response and environmental response, thus expressing the relationship between ecosystem and human activities. Some scholars have comprehensively evaluated the potential hazards of ecological security and sustainable development in the Beijing–Tianjin–Hebei mountainous area based on the fragmentation of ecological environment patches and the threat of mountain floods [30]. The existing research has laid a strong foundation for land ecological security evaluation and optimization regulation in the Beijing–Tianjin–Hebei region. In this paper, based on the transformation of major social contradictions in China, under the background of vigorously promoting ecological civilization construction, green development and high-quality development strategy, people's ever-growing needs for a better life have changed, and land ecological pressures, land ecological status and land ecological construction responses have changed in terms of connotation, standards and objectives. The factors influencing and restricting land ecological security are also different. Therefore, this paper uses the PSR model and an entropy-weight TOPSIS method to build a land ecological security evaluation index system, evaluate and analyze the level and spatial-temporal pattern of land ecological security in the Beijing–Tianjin–Hebei region. Additionally, this paper uses the obstacle degree model to identify the impact factors and main obstacle factors affecting land ecological security in the Beijing–Tianjin–Hebei region. It provides the basis for the optimization and control of land ecological security pattern and ecological construction management in the Beijing–Tianjin–Hebei region and provides theoretical reference for future land resource planning and management and the optimization of land space development and protection pattern.

## 2. Research Methods and Data Sources

### 2.1. Construction of Evaluation Index System

Land ecological security refers to the way in which the land ecosystem maintains the stability and virtuous cycle of the land system, resists land environmental risks caused by natural causes and human factors and provides a sufficient quality and quantity of services for human survival and development. Land ecological security is the basis of regional ecological security, which is related to the sustainable development of the region. The limit of the internal endurance of the land ecosystem is called the reasonable threshold of the land ecosystem. When the external disturbance exceeds this limit, the balance of the land ecosystem will be broken. The pressure–state–response (PSR) model is a framework system used to study environmental problems, which can consider many factors such as population, environment, economy and resources, and to analyze and study the comprehensive factors affecting land ecological security from the overall perspective of the system. As a subsystem in the frame of the PSR model, human activity is the driving force or pressure of the whole system, and the state of the said system, as well as resources and environment condition management, decision making and policy response to the state, can reflect the land ecological system pressure, state and response to the new connotation and target applicable to the land ecological security evaluation. Therefore, this paper comprehensively considers social, economic, resource and environmental factors, integrates the bearing pressure of the land ecosystem, the state of land resources and environment, and the response degree of ecological construction and environmental protection, and uses the PSR model framework to construct the evaluation index system of land ecological security, as shown in Table 1.

**Table 1.** Evaluation index system and index weight of land ecological security in The Beijing–Tianjin–Hebei region.

| Criterion Layer | Factor Layer | Index Layer | Direction | Weight | | |
|---|---|---|---|---|---|---|
| | | | | 2010 | 2015 | 2020 |
| Pressure | Population pressure | $P_1$ Population density (per/km$^2$) | − | 0.0215 | 0.0208 | 0.0205 |
| | | $P_2$ Natural population growth rate (‰) | − | 0.0392 | 0.0381 | 0.0238 |
| | Environmental pressure | $P_3$ Amount of fertilizer applied per unit cultivated area (kg/hm$^2$) | − | 0.0213 | 0.0207 | 0.0208 |
| | | $P_4$ Land average industrial waste discharge (t/km$^2$) | − | 0.0179 | 0.0199 | 0.0110 |
| | Economic pressure | $P_5$ Economic density (Ten thousand yuan/km$^2$) | − | 0.0180 | 0.0174 | 0.0116 |
| | | $P_6$ GDP energy consumption (10 thousand yuan/ ton standard coal) | − | 0.0218 | 0.0254 | 0.0236 |
| State | Resource environment state | $S_1$ Per capita cultivated land area (m$^2$/per) | + | 0.0352 | 0.0338 | 0.0393 |
| | | $S_2$ Per capita forestland area (m$^2$/per) | + | 0.1443 | 0.1418 | 0.1449 |
| | | $S_3$ Grassland area per capita (m$^2$/per) | + | 0.1237 | 0.1213 | 0.1260 |
| | | $S_4$ Water area per capita water area (m$^2$/per) | + | 0.0641 | 0.0648 | 0.0555 |
| | Social economic state | $S_5$ Urbanization rate (%) | + | 0.1093 | 0.1061 | 0.0700 |
| | | $S_6$ GDP per capita (yuan/per) | + | 0.0597 | 0.0579 | 0.0699 |
| | | $S_7$ Per capita disposable income (yuan) | + | 0.0252 | 0.0244 | 0.0827 |
| Response | Social economic response | $R_1$ The proportion of output value of tertiary industry in GDP (%) | + | 0.0566 | 0.0550 | 0.0303 |
| | | $R_2$ Household garbage disposal rate (%) | + | 0.0098 | 0.0095 | 0.0096 |
| | | $R_3$ Sewage treatment rate (%) | + | 0.0246 | 0.0182 | 0.0180 |
| | | $R_4$ General industrial solid waste comprehensive utilization rate (%) | + | 0.0151 | 0.0120 | 0.0136 |
| | Ecological environment response | $R_5$ Afforestation area (hm$^2$) | + | 0.0571 | 0.0554 | 0.0431 |
| | | $R_6$ Park green area (hm$^2$) | + | 0.1009 | 0.1219 | 0.1417 |
| | | $R_7$ Green rate of built-up area (%) | + | 0.0149 | 0.0145 | 0.0286 |
| | | Forest coverage rate (%) | + | 0.0197 | 0.0209 | 0.0156 |

Note: "+" and "−" represent the positive and negative indicators, respectively. "Yuan": "Yuan" is the unit of measurement for RMB (China's legal currency).

Land ecological pressure mainly comes from the impact of human production and life, which is reflected in the impact of the scope and intensity of human activity on land ecological change, and can be decomposed into population pressure, environmental pressure and economic pressure. Among these, the population pressure refers to the connotation of land ecological safety and is the guarantee of the existing material conditions, of the ability to accommodate a certain population and of the provision of food for humans. Population pressure also refers to the present situation of the population on the land ecological system and the influence of the future population, often through population density (P$_1$) and the natural population growth rate (P$_2$) [31]. The greater the population density, the greater the population per unit area of land and the greater the demand for land rearing and land carrying functions, which means the greater land use intensity. The higher the natural population growth rate, the greater the new demand for land. Environmental stress refers to the ecological system of the land's ability to prevent the effects of ecological degrada-

tion and environmental pollution from the economy, maintain the ecological system in a virtuous cycle, avoid waste material produced by human production and existence on the land's ecological environment. Rural areas that can be cultivated by unit type ($P_3$) [32] reflect the level of fertilizer. The pressure on cultivated land resources is caused by the irrational application of chemical fertilizers. Urban areas can be characterized by land average industrial waste discharge ($P_4$), which represents the pressure on land resources. Excessive fertilization and industrial waste gas, wastewater and waste residue will cause environmental pollution and threaten the security of the land ecosystem. Economic pressure refers to the impact of human economic activities on the destruction of the land ecosystem. The greater economic density ($P_5$) and GDP energy consumption ($P_6$), the greater the efficiency of urban economic activities and the intensity of land use, and the greater the intensity of human activities on land, which easily causes land ecological security problems [33].

Land ecological state refers to the state of land resources under pressure from human activities and the natural state, including the state of resources and the environment and the state of the social economy. The state of resources and the environment is mainly reflected in the quantity of resources. In combination with the land use status and ecosystem distribution in Beijing–Tianjin–Hebei, the per capita land use area can be selected to measure the land status [34]. Arable land is not only productive, but also serves as an ecological service. The 2030 Agenda for Sustainable Development refers to "protecting, restoring and promoting the sustainable use of terrestrial ecosystems, sustainably managing forests, combating desertification, halting and reversing land degradation and curbing the loss of biodiversity". Woodland, grassland and water area play an important ecosystem service function and play an important role in maintaining the balance of land ecosystem. Therefore, the per capita level of the four types of land area is selected to represent the relative level of land resources and environment state. The socioeconomic status focuses on the land use mode and land output capacity. The urbanization rate ($S_5$) represents the population flow status, reflecting the urban population and the scale of urban construction land. The higher the urbanization rate, the higher the economic benefits, representing a higher level of development. Per capita GDP ($S_6$) and per capita disposable income ($S_7$) respectively represent the level of economic development and residents' living standards. The larger the index value, the better the economic development.

Land ecological response represents the feedback measures taken by human beings when they are faced with land ecological degradation, environmental damage and ecological security, which generally includes two aspects: socio-economic response and eco-environmental response. The 2030 Agenda for Sustainable Development seeks "to reduce the number of deaths and diseases caused by air, water and soil pollution" and "to combat desertification and restore degraded lands and soils, including those affected by desertification, drought and flooding, towards a world where land degradation is no longer an issue". From the perspective of land socio-economic response, a series of measures should be taken to alleviate the land ecological security situation, protect the land ecosystem and optimize the regulation of social economic activities. Common measures include industrial structure optimization and adjustment, green production, and the development of a circular and low-carbon economy. The proportion of tertiary industrial output to GDP ($R_1$) reflects the level of industrial development and the developmental ability of the green and low-carbon economy. Tertiary industry has the advantages of low resource consumption and awareness of pollution. The higher the proportion of tertiary industry output, the greater the contribution of economic response to ecological security. The domestic waste treatment rate ($R_2$) and sewage treatment rate ($R_3$) represent the response of government policies and economic input to land ecological security and reflect the level of ecological environment management. The comprehensive utilization rate of general industrial solid waste ($R_4$) indicates that the four indicators of ecosystem protection and environmental pollution control ability can comprehensively represent the social and economic response. Ecological environment response reflects ecological construction action measures while afforestation area ($R_5$), park green space areas ($R_6$), green land rate ($R_7$) and forest coverage rate ($R_8$)

represent regional greening degree [35]. The above indicators reflect the ecological benefits brought by forests and the quantity, quality and level of urban greening. The higher the regional greening degree, the higher the land ecological security level.

*2.2. Calculation Method*

2.2.1. Land Ecological Security Entropy-Weight TOPSIS Evaluation Method

Entropy-weight method is an objective method to determine the weight according to the index variability, which is used for the comprehensive evaluation of objects. In general, the larger the entropy value, the smaller the difference coefficient and the smaller the index weight. On the contrary, the index weight is greater. The technique for order preference by similarity to ideal solution (TOPSIS) is a commonly used decision analysis method based on the principle of approximate ideal solution ranking method. It is a common evaluation method when evaluation indexes have multiple objectives [36]. By combining the entropy-weight method and TOPSIS method, the measurement results are more objective and reasonable [37], something which has been applied well in the evaluation of land use and high-quality development [38,39]. The steps and formulas of entropy-weight TOPSIS evaluation method are as follows:

Firstly, the entropy-weight method is used to determine the index weight:

$$w_j = (1 - H_i) / \sum_{i=1}^{m} (1 - H_i). \tag{1}$$

Secondly, the TOPSIS method is used to calculate the closeness degree between the evaluation object and the optimal scheme, and the positive and negative ideal solutions are determined. $D_i^+$ represents the most preferred scheme, $D_i^-$ represents the least preferred scheme, and the distance between the evaluation object and the optimal scheme is calculated.

$$D_i^+ = \max(r_{1j}, r_{2j}, \ldots, r_{nj}), \ D_i^- = \min(r_{1j}, r_{2j}, \ldots, r_{nj})$$

$$C_i^- = \sqrt{\sum_{j=1}^{n} \left(D_j^- - r_{ij}\right)^2} \tag{2}$$

$$C_i = C_i^- / C_i^+ + C_i^-, \ C_i \in [0, 1]$$

In Formula (1), $H_i$ represents the entropy of the $i$th evaluation index, $m$ represents the number of index items, and $w_j$ represents the entropy weight of index $j$. In Formula (2), $C_i$ represents the comprehensive evaluation index of land ecological security. The larger $C_i$ is, the higher the level of land ecological security is.

2.2.2. Obstacle Model of Land Ecological Security

To regulate and manage land ecological security, it is necessary to further analyze the influencing factors of land ecological security. The obstacle degree model can identify the obstacle factors affecting decision-making goals and has been applied in the analysis of obstacle factors of land ecological security and ecological construction level [40,41]. In this paper, the obstacle degree model is used to calculate and identify the main obstacle factors of land ecological security system. The specific calculation is as follows:

$$Z_{ij} = \frac{(1 - b_{ij}) w_{ij}}{\sum_{i=1}^{n} (1 - b_{ij}) w_{ij}} * 100\% \tag{3}$$

$$Z_i = \sum Z_{ij}$$

where, $Z_{ij}$ represents the obstacle degree of each index to land ecological security, $b_{ij}$ represents the standardized value of the $j$th index of the $i$th city, $w_{ij}$ represents the corresponding weight of the index, $n$ represents the number of indicators, and $Z_i$ represents the obstacle degree of the $i$th dimension to land ecological security.

### 2.2.3. Evaluation Standard of Land Ecological Security Level

The land ecological security index calculated according to the entropy-weight TOPSIS model ranges from 0 to 1, and the closer the value is to 1, the higher the land ecological security degree is. In order to deeply analyze the degree of land ecological security and explore the regional spatial pattern of security level, based on the existing research results [42,43], combined with the characteristics of the land ecosystem in the Beijing–Tianjin–Hebei region, it is divided it into five levels according to the unequal spacing classification method. These levels are safe, relatively safe, critically safe, unsafe and extremely unsafe, and are shown in Table 2.

**Table 2.** Evaluation standard of land ecological security level in the Beijing–Tianjin–Hebei region.

| Degree of Safety | Safe | Relatively Safe | Critically Safe | Unsafe | Extremely Unsafe |
|---|---|---|---|---|---|
| Grade | I | II | III | IV | V |
| Evaluation interval | [0.7, 1] | [0.5, 0.7) | [0.3, 0.5) | [0.2, 0.3) | [0, 0.2) |

### 2.3. Study Area and Data Sources

The Beijing–Tianjin–Hebei region is in North China, between 113°27′–119°50′ east longitude and 36°05′–42°40′ north latitude. The geographical location and land use types are shown in Figure 1. The Beijing–Tianjin–Hebei region is the economic center of northern China, accounting for 8.5% of China's GDP in 2020. The Beijing–Tianjin–Hebei region includes Beijing, Tianjin and 11 cities in Hebei province—Shijiazhuang, Baoding, Zhangjiakou, Tangshan, Langfang, Handan, Qinhuangdao, Chengde, Cangzhou, Xingtai and Hengshui. The Beijing–Tianjin–Hebei region is rich in topography and landforms, consisting of Bashang Plateau, Taihang Mountain, Yanshan Mountain, the North China Plain and the Bohai Plain. The plains of Beijing, Tianjin and Hebei account for about 38%, 93% and 39.8% of their respective areas. The ecosystem of the Beijing–Tianjin–Hebei region is relatively fragile. The forest coverage rate of Beijing is 44%, that of Hebei province is 26.78%, and that of Tianjin is only 12.07%. The per capita area of cultivated land is far lower than the average level of Chinese cities. For a long time, the contradiction between economic society, resources, environment and ecology has been prominent, and the problems of land desertification, soil erosion and extensive land use are serious. The study of land ecological security in the Beijing–Tianjin–Hebei region is of great significance.

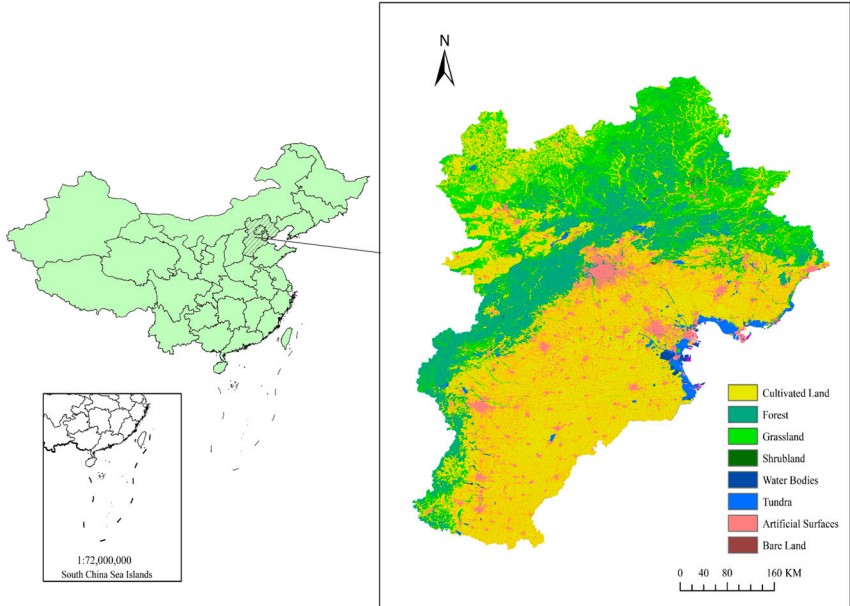

**Figure 1.** Study area geographical location and land use type distribution map.

For a long time, the contradiction between economy and society, resources, environment and ecology have been prominent, and the problems of land desertification, soil erosion and extensive land use are relatively serious. In the past 30 years, the Beijing–Tianjin–Hebei region has strengthened the restoration of the land ecosystem, especially the systematic restoration and comprehensive management of mountains, rivers, forests, fields, lakes, grass and sand in the past ten years. The land ecological security situation has been improving day by day. For example, decades of sand control and desertification control in Saihanba Mechanical Forest Farm in Hebei province have changed the desert and sandy ecology of the past—where "yellow sand covers the sky and birds do not live in trees"—and built a strong green ecological barrier. In 2017, the builders of Saihanba Mechanical Forest Farm in Hebei province were awarded the "Guardian of the Earth Award" by the United Nations Environment Program, and in 2021 they won the "Land Life Award", the highest honor of the United Nations in the field of desertification prevention and control. Grassland degradation in the Bashang grassland of Zhangjiakou was caused by people's conversion of grassland into cultivated land, overgrazing, over-development of tourism and other factors. To realize the rational utilization of land, a series of measures against grassland degradation were carried out. Nowadays, the Bashang grassland in Zhangjiakou has a beautiful environment, which is called the "Grassland Heaven Road". Figure 2 shows the Bashang grassland in Zhangjiakou. Given the above, the study of the spatial and temporal patterns of land ecological security in the Beijing–Tianjin–Hebei region is of great significance.

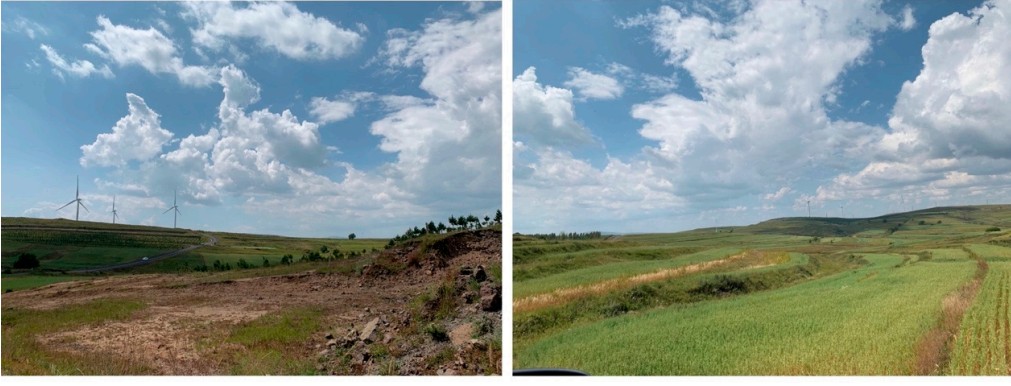

**Figure 2.** Bashang grassland of Zhangjiakou. Left picture: destruction situation. Right picture: governance situation (Photo by Liu Dandan, author).

The data of this study are derived primarily from various Chinese statistical yearbooks and bulletin data. Among these are included the National Statistical Yearbook, China Urban Statistical Yearbook and China Environmental Statistical Yearbook in 2011, 2016 and 2020, Hebei Economic Yearbook, Hebei Rural Statistical Yearbook, Beijing Statistical Yearbook and Tianjin Statistical Yearbook in 2011, 2016 and 2020, Statistical bulletins on national economic and social development of prefecture-level cities in 2011, 2016 and 2020. The data for Shijiazhuang include the data for Xinji, and the data for Baoding include the data for Dingzhou.

## 3. Results and Analysis

### 3.1. Spatial-Temporal Pattern Analysis of Land Ecological Security

This paper studied and applied an entropy-weight TOPSIS model of Equations (1) and (2). Taking 13 prefecture-level cities in Beijing, Tianjin and Hebei province as units, it calculated the land ecological security index of the Beijing–Tianjin–Hebei region at three time points in 2010, 2015 and 2019, and classified it according to Table 2. The spatial pattern of urban land ecological security in Beijing–Tianjin–Hebei is shown in Figure 3, and the ecological security index of each urban subsystem is shown in Figure 4.

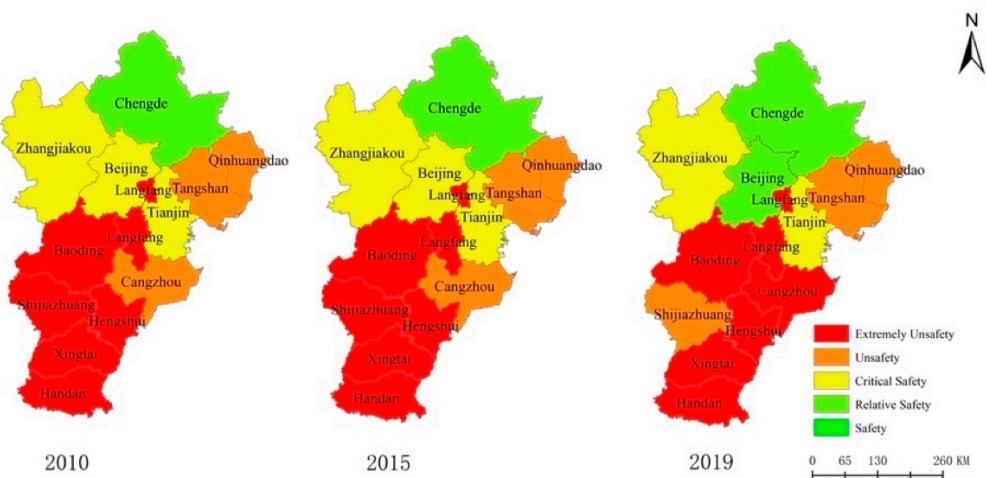

**Figure 3.** Spatial pattern of land ecological security in The Beijing–Tianjin–Hebei region.

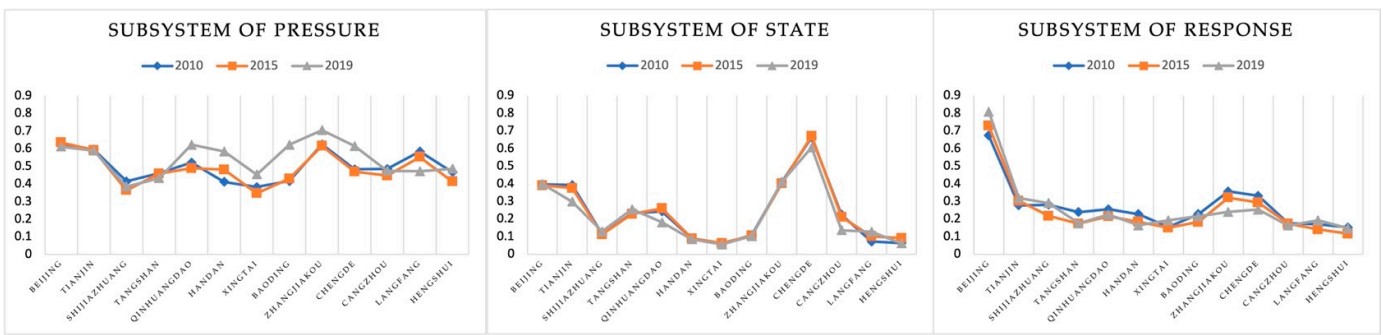

**Figure 4.** Ecological security index of urban land ecological security subsystem in the Beijing–Tianjin–Hebei region.

### 3.1.1. Current Pattern of Land Ecological Security

Land ecological security in the Beijing–Tianjin–Hebei region shows a pattern of "higher in the north and lower in the south", with a low overall security level. The northern region is relatively safe, while the southern region is insecure. The ecological security level of Chengde and Beijing is good and in the relatively safe state (II), while the ecological security levels of Zhangjiakou and Tianjin are relatively good and in the critically safe state (III). The ecological security levels of Qinhuangdao, Tangshan and Shijiazhuang are in the unsafe state (IV). Five cities, Cangzhou, Langfang, Baoding, Hengshui, Xingtai and Handan, were in the extremely unsafe state (V). From 2010 to 2019, the number of cities in the safe state (I) increased. In 2010, only one city was in the safe state (I), and in 2019, two cities were in the safe state (I). The ecological security state of Beijing changed from the critically safe state (III) to the safe state (I). According to the evaluation index of land ecological security in the Beijing–Tianjin–Hebei region, the overall index showed a downward trend, indicating that the land ecological security in the Beijing–Tianjin–Hebei region had a worsening trend. The land ecological security status in Langfang, Xingtai, Shijiazhuang and Baoding did not change, but the land ecological security index showed a fluctuating upward trend. In Tianjin, Tangshan, Qinhuangdao, Handan, Zhangjiakou, Chengde, Cangzhou and Hengshui, the land ecological security index decreased.

According to the calculation of the index values of the ecological security subsystem in Figure 4, in the pressure subsystem, Beijing, Tianjin, Zhangjiakou, Qinhuangdao and Langfang are in the safe state, while the other prefectural-level cities are in the general safe state. In the state subsystem, Chengde City is in the safe state, Beijing, Tianjin and Zhangjiakou are in the critically safe state, and the rest of the prefecture-level cities are in the extremely unsafe state. In the response subsystem, Beijing's response security state changed from critically safe in 2010 to safe in 2015 and 2019. Land ecological security level is mainly

affected by state subsystem and response subsystem. In contrast, the average index of the land ecological security pressure subsystem is the highest at 0.543, reaching the safe state and indicating that the coordinated development of the Beijing–Tianjin–Hebei region and the dispersal of non-capital functions in Beijing have increasingly coordinated population, resources, environment and the economy, and effectively alleviated ecological pressure.

Zhangjiakou is the safest place to expand the pressure subsystem, with a closeness degree of 0.706. The ecological pressure security status of Zhangjiakou, Qinhuangdao, Handan, Chengde and Baoding gradually improved in 2019, and the ecological pressure gradually decreased. Security status for Beijing, Tianjin and Cangzhou saw a slight drop, indicating that the subsystem security level is relatively poor. The Chengde state subsystem index value ranks first, keeping a good state of ecological security. The ecological security statuses of Zhangjiakou, Beijing and Tianjin are relatively stable, providing support for the overall ecological security level. The ecological security levels of Handan, Hengshui and Xingtai are low, and have a certain degree of influence on the overall ecological security level of the city. Beijing has the highest level of response subsystem, which indicates that Beijing has taken active actions in the construction of land ecological security and achieved remarkable results in the optimization of industrial structures, the treatment of "three wastes" and afforestation. The lowest levels are for Hengshui, Cangzhou, Handan and Tangshan, these cities "three waste" treatment and comprehensive utilization, afforestation area and green area are low, indicating that ecological construction is seriously lagging.

3.1.2. Temporal Change of Land Ecological Security

The land ecological security index in the Beijing–Tianjin–Hebei region has decreased slightly since 2010, mainly because the status subsystem and response subsystem indexes decreased slightly from 0.236 and 0.271 in 2010 to 0.219 and 0.260, respectively. The main influencing factors are urbanization rate, afforestation area, per capita cultivated land area, per capita grassland area, and forest coverage rate. The improvement of urbanization level is often accompanied by the expansion of construction land and the reduction of cultivated land, coupled with population growth and insufficient afforestation area, and the growth of forest coverage rate and green land rate is relatively slow. From the level of prefectures, the land ecological security index of Tianjin, Qinhuangdao, Handan, Zhangjiakou, Chengde, Cangzhou and Hengshui all declined to different degrees, and the index values of Shijiazhuang, Tangshan, Xingtai and Baoding fluctuated, showing a trend of first declining and then rising. For example, from 2010 to 2019, the urbanization rate of Tianjin increased from 79.60% to 83.50%, and the cultivated land area per 10,000 people decreased from 307.01 hectares to 227.04 hectares, which affected the level of land ecological security to some extent.

From the analysis of the land ecological security subsystem, the mean values of the pressure subsystem index in 2010, 2015 and 2019 were 0.498, 0.484 and 0.542, respectively, showing a downward trend and then an upward trend. The mean values of the state subsystem index were 0.236, 0.239, 0.219, respectively, showing a trend of increasing first and then decreasing. The mean values of the response subsystem index were 0.271, 0.246 and 0.260, respectively, showing a downward trend and then an upward trend. From the perspective of land ecological security level change, the time change of land ecological security in the Beijing–Tianjin–Hebei region has three states. Firstly, the land ecological security level improved in Beijing and Shijiazhuang. Beijing has changed from the critically safe state (III) in 2010 and 2015 to the relative safe state (II) in 2019. The ecological security pressure has been alleviated and the ecological environment has been improving. The security level of Shijiazhuang changed from the extremely unsafe state (V) in 2010 and 2015 to the unsafe state (IV) in 2019, indicating that the ecological security system of Shijiazhuang urgently needs to be improved, and the ecological security pressure, ecological security state and ecological security response should be strengthened. Second, the land ecological security levels were unchanged for Chengde, Tianjin, Zhangjiakou, Tangshan, Qinhuangdao, Handan, Xingtai, Baoding, Langfang and Hengshui. Among

these, Handan, Xingtai, Baoding, Langfang and Hengshui are in the extremely unsafe state (V). Such cities account for the majority, indicating that the construction of land ecological security in the Beijing–Tianjin–Hebei region has not achieved outstanding results. Third, the status of land ecological security deteriorated only in Cangzhou, the security status changed from the unsafe state (IV) in 2010 and 2015 to the extremely unsafe state (V) in 2019. From 2010 to 2019, the population of Cangzhou increased rapidly and the urbanization rate did not change much. However, the forest coverage rate decreased from 14.9% to 9.7%, the cultivated land area per 10,000 people decreased from 970.39 hectares to 963.50 hectares, and the forestland area per 10,000 people decreased from 286.09 hectares to 176.88 hectares, indicating poor coordination between population and resources and environment and that the problem of land ecological security is a concern.

### 3.2. Obstacle Factors of Land Ecological Security

For obstacle factor analysis, this paper adopts the disorder degree model to analyze the contribution of introduced factors, for index deviation degree, and for diagnosis of the degree of the Beijing–Tianjin–Hebei region's land ecological security barrier factor. The Beijing–Tianjin–Hebei region is calculated via formula (3), with 13 cities as the obstacle factors of obstacle degree. We determine the impact of land ecological security index, and the obstacle factors for each of the prefecture-level cities of the Beijing–Tianjin–Hebei region. Due to the large number of indicators involved, the top three obstacle factors were selected as the main obstacle factors for analysis and diagnosis for the convenience of analysis, as shown in Table 3.

**Table 3.** Obstacle factors and degree of land ecological security index layer in the Beijing–Tianjin–Hebei region.

| City | Year | 2010 | | | 2015 | | | 2019 | | |
|---|---|---|---|---|---|---|---|---|---|---|
| | Sort | 1 | 2 | 3 | 1 | 2 | 3 | 1 | 2 | 3 |
| Beijing | Obstacle factors | $P_5$ | $P_1$ | $R_3$ | $P_5$ | $P_1$ | $R_3$ | $P_5$ | $R_3$ | $P_1$ |
| | Degree | 28% | 17% | 15% | 28% | 17% | 13% | 38% | 19% | 17% |
| Tianjin | Obstacle factors | $P_5$ | $P_4$ | $P_1$ | $P_5$ | $P_4$ | $P_1$ | $P_4$ | $P_5$ | $R_3$ |
| | Degree | 27% | 18% | 16% | 27% | 17% | 16% | 33% | 21% | 19% |
| Shijiazhuang | Obstacle factors | $P_3$ | $P_4$ | $S_4$ | $P_4$ | $P_3$ | $S_4$ | $P_3$ | $P_2$ | $S_4$ |
| | Degree | 17% | 13% | 12% | 18% | 17% | 11% | 18% | 12% | 11% |
| Tangshan | Obstacle factors | $P_6$ | $P_4$ | $P_3$ | $P_6$ | $R_4$ | $P_3$ | $P_4$ | $P_6$ | $P_3$ |
| | Degree | 17% | 14% | 11% | 14% | 13% | 11% | 15% | 14% | 12% |
| Qinhuangdao | Obstacle factors | $R_4$ | $P_3$ | $S_7$ | $R_4$ | $P_3$ | $S_7$ | $P_3$ | $R_3$ | $S_1$ |
| | Degree | 13% | 11% | 10% | 15% | 11% | 10% | 10% | 9% | 9% |
| Handan | Obstacle factors | $P_3$ | $P_6$ | $S_4$ | $P_3$ | $S_4$ | $R_8$ | $P_3$ | $S_4$ | $R_7$ |
| | Degree | 15% | 14% | 13% | 15% | 12% | 10% | 13% | 11% | 11% |
| Xingtai | Obstacle factors | $S_4$ | $P_2$ | $R_7$ | $S_4$ | $P_2$ | $R_7$ | $P_2$ | $P_6$ | $S_4$ |
| | Degree | 13% | 12% | 12% | 12% | 12% | 12% | 12% | 11% | 11% |
| Baoding | Obstacle factors | $R_7$ | $P2$ | $S_6$ | $R_3$ | $R_7$ | $P_2$ | $R_2$ | $R_3$ | $S_5$ |
| | Degree | 14% | 12% | 11% | 20% | 14% | 12% | 92% | 25% | 10% |
| Zhangjiakou | Obstacle factors | $R_2$ | $R_4$ | $R_7$ | $R_7$ | $R_4$ | $P_6$ | $R_4$ | $R_7$ | $P_6$ |
| | Degree | 69% | 22% | 20% | 20% | 20% | 10% | 16% | 12% | 10% |
| Chengde | Obstacle factors | $R_4$ | $P_2$ | $P_6$ | $R_4$ | $R_2$ | $P_2$ | $R_4$ | $P_6$ | $S_5$ |
| | Degree | 29% | 13% | 10% | 35% | 17% | 13% | 23% | 11% | 11% |
| Cangzhou | Obstacle factors | $S_7$ | $P_2$ | $R_3$ | $S_7$ | $P_2$ | $S_5$ | $P_2$ | $R_5$ | $R_8$ |
| | Degree | 13% | 11% | 11% | 13% | 11% | 10% | 17% | 12% | 10% |
| Langfang | Obstacle factors | $R_2$ | $S_4$ | $S_7$ | $R_2$ | $S_4$ | $S_7$ | $P_2$ | $R_5$ | $S_4$ |
| | Degree | 17% | 13% | 11% | 65% | 12% | 11% | 17% | 12% | 11% |
| Hengshui | Obstacle factors | $S_4$ | $S_6$ | $P_2$ | $R_3$ | $S_4$ | $S_6$ | $R_5$ | $P_3$ | $S_4$ |
| | Degree | 13% | 11% | 11% | 22% | 12% | 11% | 12% | 12% | 11% |

As can be seen from Table 3, the top three obstacle factors in the Beijing–Tianjin–Hebei region remained basically unchanged from 2010 to 2019. The occurrence times of obstacle

factors in 2010, 2015 and 2019 were summarized. The top three obstacle factors in 2010 were urbanization rate ($S_5$), afforestation area ($R_5$) and sewage treatment rate ($R_3$). Urbanization was the main factor restricting the development of land ecological security, and economic development caused the pressure of land ecological security development. In 2015, the top three barriers were per capita grassland area ($S_3$), urbanization rate ($S_5$) and afforestation area ($R_5$). 2019 times the obstacle of the top three factors for the grass area ($S_3$), per capita forest land area per capita ($S_2$), established the rate ($R_7$), built up area green space area and ecological benefit brought by the afforestation area cannot compensate for the negative pressure, economic development to become main obstacle restricting the development of Beijing-Tianjin-Hebei ecological security status factors, on the whole, The unstable level of urbanization, excessive economic pressure and insufficient per capita amount of various land use areas in the Beijing–Tianjin–Hebei region are the main obstacles to the low level of land ecological security in the region. As can be seen from the obstacle factors, most of the obstacle factors that appear more often come from the state subsystem and the response subsystem, which indicates that these two subsystems are the short board sub-systems to improve the level of land ecological security in The Beijing–Tianjin–Hebei region.

From the analysis of different cities, the cities with a higher land ecological security index than the average, including Chengde, Beijing, Zhangjiakou and Tianjin, still have some prominent obstacles, and it is necessary to focus on solving these shortcomings to enhance land ecological security. For example, according to the obstacle factors in 2019, the main obstacle factor in Beijing was economic density ($P_5$), with an obstacle degree of 38%. The main obstacle factor in Tianjin is the land average industrial waste discharge ($P_4$) with an obstacle degree of 33%. The main obstacle factor in Chengde is the comprehensive utilization rate of industrial solid waste ($R_4$), which is 23%. The cities with land ecological security index lower than the average value, including Qinhuangdao, Tangshan, Cangzhou, Shijiazhuang, Baoding, Handan, Langfang, Hengshui, Xingtai, have many obstacle factors, and the obstacle degree of each factor is not too high. It is necessary to comprehensively strengthen the construction of land ecological security. Handan, Xingtai and Hengshui are the prefecture-level cities with strong restrictions on land ecological security in the Beijing–Tianjin–Hebei region. The main obstacle factors of Handan city are fertilizer application rate per unit cultivated land area ($P_3$), water area per capita ($S_4$) and green land rate in built-up area ($R_7$). The main obstacle factors of Xingtai are natural population growth rate ($P_2$), energy consumption per ten-thousand-yuan GDP ($P_6$) and water area per capita ($S_4$). The main obstacle factors in Hengshui are afforestation area ($R_5$), fertilizer application rate per unit cultivated land area ($P_3$) and water area per capita ($S_4$). We should strive to improve the level of land ecological pressure, land ecological state and land ecological response, and comprehensively maintain the land ecological system.

## 4. Discussion

Land ecological security is directly related to the safety and health of human existence and is the most fundamental material basis for the sustainable development of social economy. Land is a complex of nature, economy, society and environment. Land use is a complex system in which history and the present situation are intertwined and have an important impact on the future. The land ecological security concept is rich, can prevent shortage of resources, soil environmental pollution, natural disaster and other aspects of risk, and promote the stable recovery ability of the land ecological system. It is also needed to provide the adequate quantity and quality of necessities for human survival and development of land ecosystem services and to realize the harmonious coexistence between man and nature of land ecological basis. From the perspective of land ecological security and its prevention, the evaluation of land ecological security should first measure the ecological pressure and evaluate whether it exceeds the threshold of the land ecological system's carrying capacity. Secondly, we should measure the state of the ecosystem, evaluate the distance between it and the level of ecological security, and judge whether a benign cycle system within the land ecosystem has been constructed. Thirdly, we should consider the

responsibility to cope with ecological security problems and assess whether ecological security risks are effectively prevented and controlled. Therefore, with the help of the "pressure–state–response" (PSR) model and the scientific, regional and operable principles, this paper selected indicators from the aspects of economy, society and environment to construct an indicator system, used the entropy-weight TOPSIS model to evaluate land ecological security and used the obstacle degree model to identify obstacle factors. The pattern and obstacle factors of land ecological security in Beijing, Tianjin and Hebei were analyzed. The research conclusions and implications are as follows:

(1) The overall ecosystem health in the Beijing–Tianjin–Hebei region is poor [44], and the overall level of land ecological security is low. Urbanization rate, economic density and the impact of various land use areas are the main factors affecting the land ecological security level of Beijing–Tianjin–Hebei [32]. According to the data of the National Bureau of Statistics of China, the urbanization rate of the Beijing–Tianjin–Hebei region was 55.7% in 2010 and increased to 64.9% in 2017, among which Beijing and Tianjin were both above 80%. Urbanization is a new direction of economic and social development, and within a reasonable range, it is conducive to the good development of ecological security level. The level of urbanization in Beijing-Tianjin-Hebei is in a slightly unbalanced state [45]. The substantial increase of urbanization level will bring a large number of population inflows, and the high population density will bring pressure on the land [46], which will affect the land ecological security level of Beijing–Tianjin–Hebei. Economic development has a great impact on land ecological security [23], causing pressure on resources and environment. From 2010 to 2019, the proportion of tertiary industrial output to GDP in the Beijing–Tianjin–Hebei region continued to decline, and the industrial structure was unreasonable, which had an important impact on land ecological security [46]. In addition, the ecological benefits released by different land use are also important factors affecting land ecological security [47]. The green land rate and forest coverage rate of built-up areas in the Beijing–Tianjin–Hebei region are insufficient, meaning that there is not enough ecological benefits to make up for the negative pressure brought by the economic development process, resulting in an overall low level of land ecological security in the Beijing–Tianjin–Hebei region.

Through this evaluation and analysis, we can see that the task of constructing the land ecological security system and optimizing the regional pattern in the Beijing–Tianjin–Hebei region is still very difficult and requires long-term and continuous efforts. The construction of a benign cycle for the land's ecosystem and a system of risk prevention and control of the land's ecological environment in the Beijing–Tianjin–Hebei region should be accelerated. The population distribution should be further regulated, the economic density of land should be rationally distributed, and the pressure of land population should be generally alleviated. The area of ecological land should be expanded, the protection of woodland, grassland and wetland strengthened, the land ecological quality improved, and the land ecological self-recovery ability enhanced with a continued increase of investment in pollution prevention and control. There should also be a step up of the monitoring of ecological security, a strengthening of the restoration of land ecosystems, an improvement in the pattern of land resource development and protection, and an acceleration of the formation of land ecological security.

(2) The land ecological status of the Beijing–Tianjin–Hebei region is improved in the northern mountainous plateau region, but less so in the southern plain region [48]. Most areas in Hebei province are at an unsafe level, because Hebei province relies more on the traditional economic development model [49]. For example, Handan has a high agricultural level, a large amount of fertilizer is used per unit of cultivated land, and the land ecological health level is low [50]. Although Baoding has high-quality water resources, pollution prevention and ecological restoration work is still not in place, and the ability to prevent and control environmental risks needs to be strengthened. The ecological level of land in Zhangjiakou, Beijing and Tianjin is generally safe, mainly due to the economic transition from extensive growth to intensive growth, and intensive land use has been strengthened. The implementation of land ecological engineering in Zhangjiakou

is of great significance for intensive land use and the improvement of soil conservation function [51]. Beijing has issued a series of land policies, carried out comprehensive land consolidation [52], established ecological conservation areas [53], solved land ecological problems, improved ecological environment conditions, and performed well in ecological security levels. Chengde city has the highest level of land ecological safety and security, reaching a safe state. With a good environmental foundation, Chengde has carried out soil and water conservation engineering construction [54] and established Saihanba National Forest Park. The forest coverage rate has increased, providing good ecological benefits for urban development.

Through evaluation and analysis, we can see that the pattern of land ecological security in the Beijing–Tianjin–Hebei region is "high in the north and low in the south". Combined with the specific situation of each city in the Beijing–Tianjin–Hebei region and related policy analysis, it is apparent that the Beijing–Tianjin–Hebei region should take effective measures to ensure the rational development and utilization of natural resources to improve the level of the land's ecological security and promote the protection of natural resources and ecological environment to the national policy level. There should be the construction of a benign cycle system for the land's ecosystem to ensure its vitality, the improvement of an effective prevention and control system for environmental risks, the restoration of polluted and damaged land caused by industrial activities, an easing of the pressure on land ecosystems, and the establishment of targeted measures to ensure sustainable and safe land ecosystems.

(3) The obstacle factors of land ecological security in the Beijing–Tianjin–Hebei region vary with time. From the perspective of subsystems, the obstacle factors affecting the level of land ecological security in the Beijing–Tianjin–Hebei region in 2010 mainly came from the state and response subsystems, and the obstacle factors affecting the level of land ecological security in the Beijing–Tianjin–Hebei region in 2019 mainly came from the state subsystem. Of these subsystems, the pressure subsystem is affected by indicators such as fertilizer application rate per unit of cultivated land, economic density and natural population growth rate. The status subsystem is limited by indicators such as per capita cultivated land area, per capita forest area, per capita grassland area, per capita water area [55] and urbanization rate. The response subsystem is affected by the proportion of output value of tertiary industry in GDP, afforestation area, forest coverage rate, green land rate of built-up area and other indicators. This phenomenon is mainly due to the reduction of human activity intensity, land pollution load and land ecological pressure in the Beijing–Tianjin–Hebei region [56], as well as the control of pesticide and fertilizer use, rational use of fertilizer, and a reduction of rural household waste discharge [57].

Through evaluation and analysis, we can see that the economic density of land should be rationally distributed in the Beijing–Tianjin–Hebei region to ensure adequate ecological land use and reduce the pressure of land ecological environment. There should be an acceleration of industrial restructuring, a promotion of green transformation, an upgrading of industries, and the achievement of high-quality economic development. There should also be an optimized farmland protection system to reduce soil pollution, expand the area of ecological land and improve its ecological quality.

## 5. Conclusions

Identifying the concept of land ecological security, this study used the PSR model, based on the principles of scientific nature, regional and operability, from the perspectives of economic, social and the environment, to build an evaluation index system, using an entropy-weight method to calculate index weight along with the TOPSIS method to evaluate the land ecological security level of the Beijing–Tianjin–Hebei region. Additionally, we used a degree model to identify the obstacle factors of the Beijing–Tianjin–Hebei region. By identifying the obstacle factors of the land's ecological security in Beijing–Tianjin–Hebei, the top three obstacle factors for each prefecture-level city were extracted. According to the comprehensive occurrence times, the three subsystems of pressure, state and response were

classified and studied. Among these, the pressure subsystem is affected by the application rate of chemical fertilizer per unit cultivated land, economic density, natural population growth rate and other indicators. The status subsystem is restricted by the per capita cultivated land area, per capita forestland area, per capita grassland area, per capita water area and urbanization rate. The response subsystem is affected by the ratio of output value of the tertiary industry to GDP, afforestation area, forest coverage rate, green land ratio of built-up areas and other indicators. Our conclusion indicates that the construction of the land ecological security system in the Beijing–Tianjin–Hebei region should be strengthened, the response of land ecological security should be enhanced, and the quality of land ecosystem should be improved. This study is of great significance for the promotion of the formation of a benign cycle for the land's ecosystem and the effective prevention and control of the land's ecological and environmental risks in the Beijing–Tianjin–Hebei region, as well as providing a basis for the optimization and control of the land's ecological security pattern and ecological construction management in the region. Based on the research conclusions, to effectively improve the level of land ecological security in the Beijing–Tianjin–Hebei region, future land management should start from the following points. First, there should be a rational distribution of the land's economic density and a reduction in the land's bearing pressure. Second, there should be an expansion of the area of ecological land and an improvement in the ecological quality of the land. Third, there should be an optimization of the population structure and an adjustment of the regional population pressure. Fourth, there should be a development of the tertiary industry and the active promotion of the ecological industry. Finally, we should improve work in afforestation projects and increase the area of afforestation.

**Author Contributions:** Project administration, W.P.; writing—original draft preparation, Y.S.; data curation, C.L.; writing—review and editing, D.L. All authors have read and agreed to the published version of the manuscript.

**Funding:** This research was funded by the Beijing Social Science Foundation Major project "Research on building an ecological civilization system" (19ZDA03).

**Institutional Review Board Statement:** Not applicable.

**Informed Consent Statement:** Not applicable.

**Data Availability Statement:** All data and models generated or used in the research process of this paper are presented and explained in the body of the article.

**Conflicts of Interest:** The authors declare no conflict of interest.

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
