# Peer review of "Study on Urban Land Ecological Security Pattern and Obstacle Factors in the Beijing–Tianjin–Hebei Region"

_sustainability, doi:10.3390/su15010043_

Round 1
Reviewer 1 Report
This topic is very interesting and important. I suggest this paper can be accepted after revision.
1. Regarding to the introduction section. (1) It is suggested to supplement relevant literature to introduce the research progress, practical problems and future research directions of ecological security in the Beijing-Tianjin-Hebei region. (2) Please clarify your research contributions in this section. (3) In this section, authors summarize the current evaluation methods of land ecological security level, but the advantages of this method are not highlighted in the article.
2. Regarding to the research methods and data sources, the authors introduce the construction of evaluation index system. It is suggested to supplement relevant literature to explain the purpose and evaluation significance of index selection.
3. Figure 2 shows the location map of the study area. It is suggested that the South China Sea area should be clearly indicated in the map of China.
4.Regarding to the conclusion section, the practical significance of this study and the guidance and suggestions for future ecological security work have not been pointed out.
5. It is suggested to sort out the references as a whole and unify the reference format.
Author Response
Dear Editor,
Thank you very much for giving us the opportunity to re-submit the revised manuscript “Study on Urban Land Ecological Security Pattern and Obstacle Factors in Beijing-Tianjin-Hebei Region” may be published in Sustainability as a research paper. The comments on our original manuscript were very insightful and constructive. We greatly appreciate the time and effort you and the reviewers have given to provide valuable feedback on our work. We revised and improved the manuscript according to the comments and gave detailed replies to each reviewer's comments. Attached is our revised manuscript, named " sustainability-1978394(20221205)".
We hope that the careful revision of the manuscript and our accompanying response will meet sustainability's academic publishing requirements. If you have any questions, please do not hesitate to contact me. We look forward to your early reply.
Best regards,
Wenying Peng
E-mail: cjpengwy@cueb.edu.cn

Reviewer 2 Report
• Title need to be change as the study conducted in different cities
• Abstract and literature review need to be shortened and a numerical data should be included in the abstract.
• The aim of the study should be clearly stated in the abstract and at the end of the literature review, and highlight clearly the major result of the aim at the conclusion body
• Figure 3 should be replaced by figure 2, the data of the tables better to be represented by visual graphs
• Discussion must be linked with the literature review and clearly highlight the contribution, How accurate are the results derived from the methods used? a validation on the methods are needed
• The terms of Sustainability and the Remote Sensing Technology should be should be reflected in the paper
• Most of The references are quit old, the new references are very limited, please update and discuss the cutting edge research on the topic
Author Response

(The authors gave the same response as above.)

Reviewer 3 Report
This paper focuses on a very important issue - Land Ecological Security Patterns and Obstacle Factors in China. It is well written and well structured. Methodology seems sound, and conclusions are in tune with the findings. Bibliography is adequate. The paper could be enriched with actual photos of good/bad examples in Beijing-Tianjin-Hebei region.
Author Response
Dear Editor,
Thank you very much for giving us the opportunity to re-submit the revised manuscript “Study on Urban Land Ecological Security Pattern and Obstacle Factors in Beijing-Tianjin-Hebei Region” may be published in Sustainability as a research paper. The comments on our original manuscript were very insightful and constructive. We greatly appreciate the time and effort you and the reviewers have given to provide valuable feedback on our work. We revised and improved the manuscript according to the comments and gave detailed replies to each reviewer's comments. The detailed reply to reviewer's comments is in the document "Response to Reviewer 3 Comments.docx", and the revised revised manuscript in the document"sustainability-1978394(20221207).docx".
We hope that the careful revision of the manuscript and our accompanying response will meet sustainability's academic publishing requirements. If you have any questions, please do not hesitate to contact me. We look forward to your early reply.
Best regards,
Wenying Peng
E-mail: cjpengwy@cueb.edu.cn

Round 2
Reviewer 2 Report
-Chart figure 4 have low resolution, improve the figure quality,
- Compare the results with previous studies and highlight the contributions. Explain what previous scholars and studies have found and what they have said about the topic, then highlight your contribution. The discussion section is not just to discuss your results, but also to discuss others' views and perspectives. All the Authors that you will discuss with, must be stated in the literature review section. please re-discuss the discussion section.
- full name of the the TOPSIS method and PSR model in the abstract
Author Response
Dear Editor,
Thank you very much for giving us the opportunity to re-submit the revised manuscript “Study on Urban Land Ecological Security Pattern and Obstacle Factors in Beijing-Tianjin-Hebei Region” may be published in Sustainability as a research paper. The comments on our original manuscript were very insightful and constructive. We greatly appreciate the time and effort you and the reviewers have given to provide valuable feedback on our work. We have revised and improved the manuscript according to the comments, and made detailed replies to each reviewer's comments. Please check the attachment named "Respond to reviewer 2 comments(Round 2)". In addition, the attachment of the revised version is named "sustainability-1978394(20221212)".
We hope that the careful revision of the manuscript and our accompanying response will meet sustainability's academic publishing requirements. If you have any questions, please do not hesitate to contact me. We look forward to your early reply.
